# Benchopt: Reproducible, efficient and collaborative optimization benchmarks

**Thomas Moreau**[1,*]**, Mathurin Massias**[2,*]**, Alexandre Gramfort**[1,*]**, Pierre Ablin**[3]**,**
**Pierre-Antoine Bannier, Benjamin Charlier**[4]**, Mathieu Dagréou**[1]**, Tom Dupré la Tour**[6]**,**
**Ghislain Durif**[4]**, Cassio F. Dantas**[7]**, Quentin Klopfenstein**[8]**, Johan Larsson**[9]**, En Lai**[1]**,**
**Tanguy Lefort**[4]**, Benoit Malézieux**[1]**, Badr Moufad**[2]**, Binh T. Nguyen**[10]**, Alain Rakotomamonjy**[11]**,**
**Zaccharie Ramzi**[12]**, Joseph Salmon**[4,5]**, Samuel Vaiter**[13]

[1] Université Paris-Saclay, Inria, CEA, 91120 Palaiseau, France
[2] Univ Lyon, Inria, CNRS, ENS de Lyon, UCB Lyon 1, LIP UMR 5668, F-69342, Lyon, France
[3] Université Paris-Dauphine, PSL University, CNRS, 75016, Paris, France
[4] IMAG, Univ Montpellier, CNRS, Montpellier, France    [5] Institut Universitaire de France (IUF)
[6] University of California, Berkeley, CA 94720, USA    [7] TETIS, Univ Montpellier, INRAE, Montpellier, France
[8] University of Luxembourg, LCSB, Esch-sur-Alzette, Luxembourg
[9] The Department of Statistics, Lund University    [10] LTCI, Télécom Paris, 91120 Palaiseau, France
[11] Criteo AI Lab, Paris, France    [12] ENS Ulm, CNRS, UMR 8553, Paris, France
[13] CNRS & Université Côte d'Azur, Laboratoire J.A. Dieudonné, CNRS, Nice, France

## Abstract

Numerical validation is at the core of machine learning research as it allows to assess the actual impact of new methods, and to confirm the agreement between theory and practice. Yet, the rapid development of the field poses several challenges: researchers are confronted with a profusion of methods to compare, limited transparency and consensus on best practices, as well as tedious re-implementation work. As a result, validation is often very partial, which can lead to wrong conclusions that slow down the progress of research. We propose `Benchopt`, a collaborative framework to automate, reproduce and publish optimization benchmarks in machine learning across programming languages and hardware architectures. `Benchopt` simplifies benchmarking for the community by providing an off-the-shelf tool for running, sharing and extending experiments. To demonstrate its broad usability, we showcase benchmarks on three standard learning tasks: $\ell_2$-regularized logistic regression, Lasso, and ResNet18 training for image classification. These benchmarks highlight key practical findings that give a more nuanced view of the state-of-the-art for these problems, showing that for practical evaluation, the devil is in the details. We hope that `Benchopt` will foster collaborative work in the community hence improving the reproducibility of research findings.

## 1 Introduction

Numerical experiments have become an essential part of statistics and machine learning (ML). It is now commonly accepted that every new method needs to be validated through comparisons with existing approaches on standard problems. Such validation provides insight into the method's benefits and limitations and thus adds depth to the results. While research aims at advancing knowledge and not just improving the state of the art, experiments ensure that results are reliable and support theoretical claims (Sculley et al., 2018). Practical validation also helps the ever-increasing number of ML users in applied sciences to choose the right method for their task. Performing rigorous and extensive experiments is, however, time-consuming (Raff, 2019), particularly because comparisons

36th Conference on Neural Information Processing Systems (NeurIPS 2022).

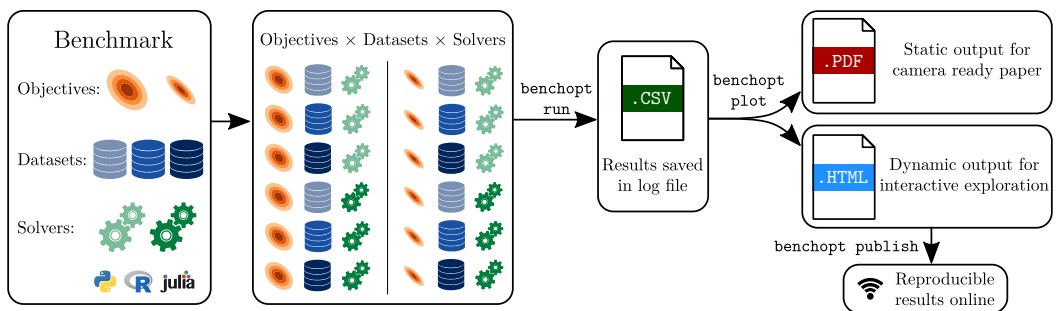

Figure 1: A visual summary of `Benchopt`. Each `Solver` is run (in parallel) on each `Dataset` and each variant of the `Objective`. Results are exported as a `CSV` file that is easily shared and can be automatically plotted as interactive `HTML` visualizations or `PDF` figures.

against existing methods in new settings often requires reimplementing baseline methods from the literature. In addition, ingredients necessary for a proper reimplementation may be missing, such as important algorithmic details, hyperparameter choices, and preprocessing steps (Pineau et al., 2019).

In the past years, the ML community has actively sought to overcome this "reproducibility crisis" (Hutson, 2018) through collaborative initiatives such as open datasets (`OpenML`, Vanschoren et al. 2013), standardized code sharing (Forde et al., 2018), benchmarks (`MLPerf`, Mattson et al. 2020), the NeurIPS and ICLR reproducibility challenges (Pineau et al., 2019; Pineau et al., 2021) and new journals (*e.g.,* Rougier and Hinsen 2018). As useful as these endeavors may be, they do not, however, fully address the problems in optimization for ML since, in this area, there are no clear community guidelines on how to perform, share, and publish benchmarks.

Optimization algorithms pervade almost every area of ML, from empirical risk minimization, variational inference to reinforcement learning (Sra et al., 2012). It is thus crucial to know which methods to use depending on the task and setting (Bartz-Beielstein et al., 2020). While some papers in optimization for ML provide extensive validations (Lueckmann et al., 2021), many others fall short in this regard due to lack of time and resources, and in turn feature results that are hard to reproduce by other researchers. In addition, both performance and hardware evolve over time, which eventually makes static benchmarks obsolete. An illustration of this is the recent work by Schmidt et al. (2021), which extensively evaluates the performances of 15 optimizers across 8 deep-learning tasks. While their benchmark gives an overall assessment of the considered solvers, this assessment is bound to become out-of-date if it is not updated with new solvers and new architectures. Moreover, the benchmark does not reproduce state-of-the-art results on the different datasets, potentially indicating that the considered architectures and optimizers could be improved.

> We firmly believe that this critical task of **maintaining an up-to-date benchmark in a field cannot be solved without a collective effort**. We want to empower the community to take up this challenge and **build a living, reproducible and standardized state of the art that can serve as a foundation for future research.**

`Benchopt` provides the tools to structure the optimization for machine learning (Opt-ML) community around standardized benchmarks, and to aggregate individual efforts for reproducibility and results sharing. `Benchopt` can handle algorithms written in `Python`, `R`, `Julia` or `C/C++` via binaries. It offers built-in functionalities to ease the execution of benchmarks: parallel runs, caching, and automatic results archiving. Benchmarks are meant to evolve over time, which is why `Benchopt` offers a modular structure through which a benchmark can be easily extended with new objective functions, datasets, and solvers by the addition of a single file of code.

The paper is organized as follows. We first detail the design and usage of `Benchopt`, before presenting results on three canonical problems:

- $\ell_2$-regularized logistic regression: a convex and smooth problem which is central to the evaluation of many algorithms in the Opt-ML community, and remains of high relevance for practitioners;
- the Lasso: the prototypical example of non-smooth convex problem in ML;
- training of ResNet18 architecture for image classification: a large scale non-convex deep learning problem central in the field of computer vision.

The reported benchmarks, involving dozens of implementations and datasets, shed light on the current state-of-the-art solvers for each problem, across various settings, highlighting that the best algorithm largely depends on the dataset properties (*e.g.,* size, sparsity), the hyperparameters, as well as hardware. A variety of other benchmarks (*e.g.,* MCP, TV1D, etc.) are also presented in Appendix, with the goal to facilitate contributions from the community.

By the open source and collaborative design of Benchopt (BSD 3-clause license), we aim to open the way towards community-endorsed and peer-reviewed benchmarks that will improve the tracking of progress in optimization for ML.

## 2 The Benchopt library

The Benchopt library aims to provide a standard toolset and structure to implement benchmarks for optimization in ML, where the problems depend on some input dataset $\mathcal{D}$. The considered problems are of the form

$$\theta^* \in \arg\min_{\theta \in \Theta} f(\theta; \mathcal{D}, \Lambda) \ , \tag{1}$$

where $f$ is the objective function, $\Lambda$ are its hyperparameters, and $\Theta$ is the feasible set for $\theta$. The **scope** of the library is to evaluate optimization methods in their wide sense by considering the sequence $\{\theta^t\}_t$ produced to approximate $\theta^*$. We emphasize than Benchopt does not provide a fixed set of benchmarks, but a framework to create, extend and share benchmarks on any problem of the form (1). To provide a flexible and extendable coding standard, benchmarks are defined as the association of three types of object classes:

**Objective:** It defines the function $f$ to be minimized as well as the hyperparameters $\Lambda$ or the set $\Theta$, and the metrics to track along the iterations (*e.g.,* objective value, gradient norm for smooth problems, or validation loss). Multiple metrics can be registered for each $\theta^t$.

**Datasets:** The Dataset objects provide the data $\mathcal{D}$ to be passed to the Objective class. They control how data is loaded and preprocessed. Datasets are separated from the Objective, making it easy to add new ones, provided they are coherent with the Objective.

**Solvers:** The Solver objects define how to run the algorithm. They are provided with the Objective and Dataset objects and output a sequence $\{\theta^t\}_t$. This sequence can be obtained using a single run of the method, or with multiple runs in case the method only returns its final iterate.

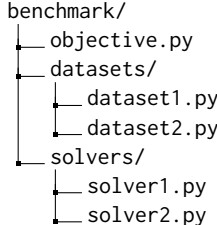

Figure 2: Standard benchmark structure

Each of these objects can have parameters that change their behavior, *e.g.,* the regularization parameters for the Objective, the choice of preprocessing for the Datasets, or the step size for the Solvers. By exposing these parameters in the different objects, Benchopt can evaluate their influence on the benchmark results. The Benchopt library defines an application programming interface (API) for each of these concepts and provides a command line interface (CLI) to make them work together. A benchmark is defined as a folder that contains an Objective as well as subfolders containing the Solvers and Datasets. Appendix B presents a concrete example on Ridge regression of how to construct a Benchopt benchmark while additional design design choices of Benchopt are discussed in Appendix C.

For each Dataset and Solver, and for each set of parameters, Benchopt retrieves a sequence $\{\theta^t\}_t$ and evaluates the metrics defined in the Objective for each $\theta^t$. To ensure fair evaluation, the computation of these metrics is done off-line. The metrics are gathered in a CSV file that can be used to display the benchmark results, either locally or as HTML files published on a website that reference the benchmarks run with Benchopt. This workflow is described in Figure 1.

This modular and standardized organization for benchmarks empowers the optimization community by making numerical experiments easily reproducible, shareable, flexible and extendable. The benchmark can be shared as a git repository or a folder containing the different definitions for the Objective, Datasets and Solvers and it can be run with the Benchopt CLI, hence becoming a convenient reference for future comparisons. This ensures fair evaluation of baselines in follow-up experiments, as implementations validated by the community are available. Moreover, benchmarks can be extended easily as one can add a Dataset or a Solver to the comparison by adding a single

file. Finally, by supporting multiple metrics – *e.g.,* training and testing losses, error on parameter estimates, sparsity of the estimate – the `Objective` class offers the flexibility to define the concurrent evaluation, which can be extended to track extra metrics on a per benchmark basis, depending on the problem at hand.

As one of the goal of `Benchopt` is to make benchmarks as simple as possible, it also provides a set of features to make them easy to develop and run. `Benchopt` is written in `Python`, but `Solvers` run with implementations in different languages (*e.g.,* `R` and `Julia`, as in Section 4) and frameworks (*e.g.,* `PyTorch` and `TensorFlow`, as in Section 5). Moreover, benchmarks can be run in parallel with checkpointing of the results, enabling large scale evaluations on many CPU or GPU nodes. `Benchopt` also makes it possible to run solvers with many different hyperparameters' values , allowing to assess their sensitivity on the method performance. Benchmark results are also automatically exported as interactive visualizations, helping with the exploration of the many different settings.

**Benchmarks** All presented benchmarks are run on 10 cores of an Intel Xeon Gold 6248 CPUs @ 2.50GHz and NVIDIA V100 GPUs (16GB). The results' interactive plots and data are available at https://benchopt.github.io/results/preprint_results.html.

## 3   First example: $\ell_2$-regularized logistic regression

Logistic regression is a very popular method for binary classification. From a design matrix $X \in \mathbb{R}^{n \times p}$ with rows $X_i$ and a vector of labels $y \in \{-1, 1\}^n$ with corresponding element $y_i$, $\ell_2$-regularized logistic regression provides a generalized linear model indexed by $\theta^* \in \mathbb{R}^p$ to discriminate the classes by solving

$$\theta^* = \underset{\theta \in \mathbb{R}^p}{\arg\min} \sum_{i=1}^{n} \log \left( 1 + \exp(-y_i X_i^\top \theta) \right) + \frac{\lambda}{2} \|\theta\|_2^2 \ , \tag{2}$$

where $\lambda > 0$ is the regularization hyperparameter. Thanks to the regularization part, Problem (2) is strongly convex with a Lipschitz gradient, and thus its solution can be estimated efficiently using many iterative optimization schemes.

The most classical methods to solve this problem take inspiration from Newton's method (Wright and Nocedal, 1999). On the one hand, quasi-Newton methods aim at approximating the Hessian of the cost function with cheap to compute operators. Among these methods, L-BFGS (Liu and Nocedal, 1989) stands out for its small memory footprint, its robustness and fast convergence in a variety of settings. On the other hand, truncated Newton methods (Dembo et al., 1982) try to directly approximate Newton's direction by using *e.g.,* the conjugate gradient method (Fletcher and Reeves, 1964) and Hessian-vector products to solve the associated linear system. Yet, these methods suffer when $n$ is large: each iteration requires a pass on the whole dataset.

In this context, methods based on stochastic estimates of the gradient have become standard (Bottou, 2010), with Stochastic Gradient Descent (SGD) as a main instance. The core idea is to use cheap and noisy estimates of the gradient (Robbins and Monro, 1951; Kiefer and Wolfowitz, 1952). While SGD generally converges either slowly due to decreasing step sizes, or to a neighborhood of the solution for constant step sizes, variance-reduced adaptations such as SAG (Schmidt et al., 2017), SAGA (Defazio et al., 2014) and SVRG (Johnson and Zhang, 2013) make it possible to solve the problem more efficiently and are often considered to be state-of-the-art for large scale problems.

Finally, methods based on coordinate descent (Bertsekas, 1999) have also been proposed to solve Problem (2). While these methods are usually less popular, they can be efficient in the context of sparse datasets, where only few samples have non-zero values for a given feature, or when accelerated on distributed systems or GPU (Dünner et al., 2018).

The code for the benchmark is available at https://github.com/benchopt/benchmark_logreg_l2/. To reflect the diversity of solvers available, we showcase a `Benchopt` benchmark with 3 datasets, 10 optimization strategies implemented in 5 packages, leveraging GPU hardware when possible. We also consider different scenarios for the objective function: (i) **scaling** (or not) the features, a recommended data preprocessing step, crucial in practice to have comparable regularization strength on all variables; (ii) fitting (or not) an unregularized **intercept** term, important in practice and making optimization harder when omitted from the regularization term (Koh et al., 2007); (iii) working (or

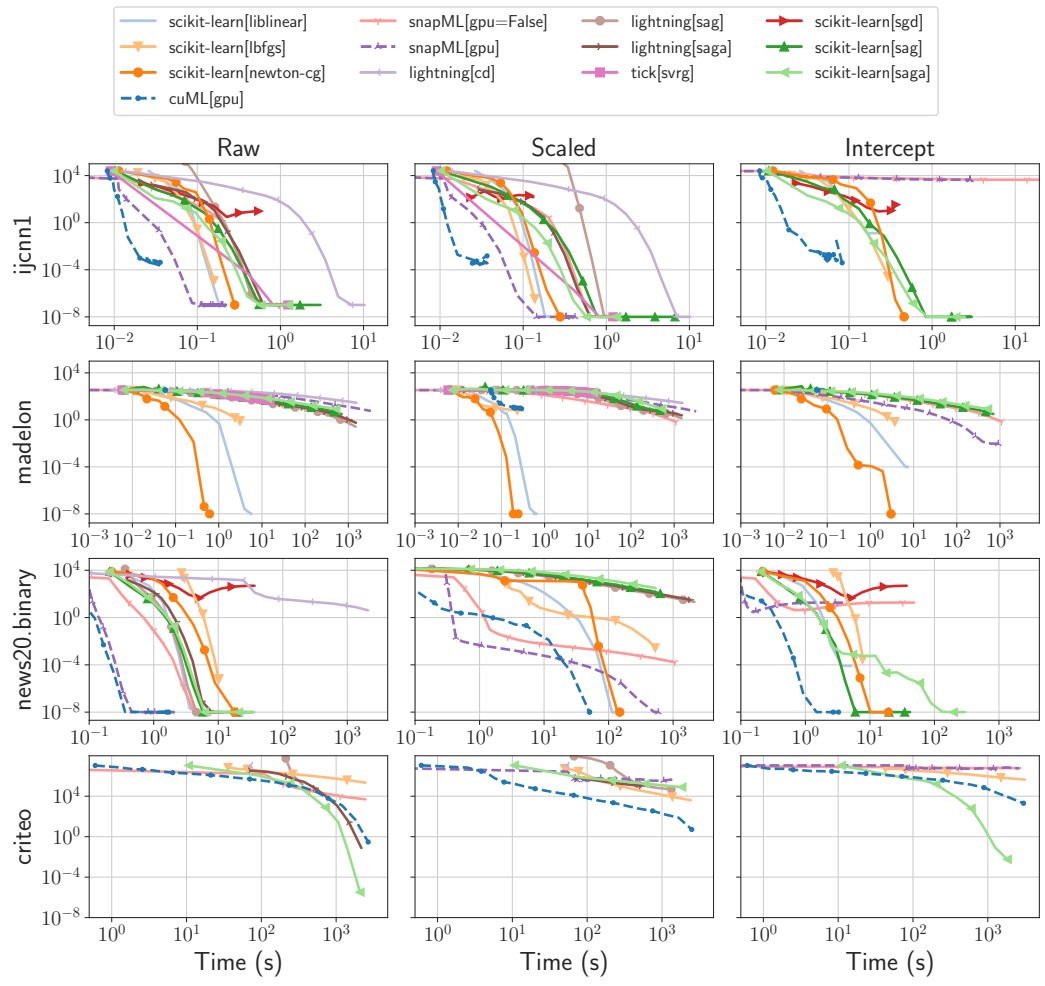

Figure 3: Benchmark for the $\ell_2$-regularized logistic regression, on 13 solvers, 4 datasets (*rows*), and 3 variants of the Objective (*columns*) with $\lambda = 1$. The curves display the suboptimality of the iterates, $f(\theta^t) - f(\theta^*)$, as a function of time. The first column corresponds to the objective function detailed in Problem (2). In the second column, datasets were preprocessed by normalizing each feature to unit standard deviation. The third column is for an objective function which includes an unregularized intercept.

not) with **sparse** features, which prevent explicit centering during preprocessing to keep memory usage limited. Details on packages, datasets and additional scenarios are available in Appendix D.

**Results** Figure D.1 presents the results of the benchmarks, in terms of suboptimality of the iterates, $f(\theta^t) - f(\theta^*)$, for three datasets and three scenarios. Here, because the problem is convex, $\theta^*$ is approximated by the best iterate across all runs (see Section C.1). Overall, the benchmark shows the benefit of using GPU solvers (cuML and snapML), even for small scale tasks such as *ijcnn1*. Note that these two accelerated solvers converge to a higher suboptimality level compared to other solvers, due to operating with 32-bit float precision. Another observation is that data scaling can drastically change the picture. In the case of *madelon*, most solvers have a hard time converging for the scaled data. For the solvers that converge, we note that the convergence time is one order of magnitude smaller with the scaled dataset compared to the raw one. This stems from the fact that in this case, the scaling improves the conditioning of the dataset.[1] For *news20.binary*, the stochastic solvers such as SAG and SAGA have degraded performances on scaled data. Here, the scaling makes the problem harder.[2]

---

[1]The condition number of the dataset is divided by 5.9 after scaling.

[2]The condition number is multiplied by 407 after scaling.

On CPU, quasi-Newton solvers are often the most efficient ones, and provide a reasonable choice in most situations. For large scale *news20.binary*, stochastic solvers such as SAG, SAGA or SVRG –that are often considered as state of the art for such problem– have worst performances for the presented datasets. While this dataset is often used as a test bed for benchmarking new stochastic solvers, we fail to see an improvement over non-stochastic ones for this experimental setup. In contrast, the last row in Figure D.1 displays an experiment with the larger scale *criteo* dataset, which demonstrates a regime where variance-reduced stochastic gradient methods outperform quasi-Newton methods. For future benchmarking of stochastic solvers, we therefore recommend using such a large dataset.

Finally, the third column in Figure D.1 illustrates a classical problem when benchmarking different solvers: their specific (and incompatible) definition and resolution of the corresponding optimization problem. Here, the objective function is modified to account for an intercept (bias) in the linear model. In most situations, this intercept is not regularized when it is fitted. However, `snapML` and `liblinear` solvers do regularize it, leading to incomparable losses.

## 4 Second example: The Lasso

The Lasso, (Tibshirani, 1996; Chen et al., 1998), is an archetype of non-smooth ML problems, whose impact on ML, statistics and signal processing in the last three decades has been considerable (Bühlmann and van de Geer, 2011; Hastie et al., 2015). It consists of solving

$$\theta^* \in \arg\min_{\theta \in \mathbb{R}^p} \tfrac{1}{2} \|y - X\theta\|^2 + \lambda \|\theta\|_1 \quad , \tag{3}$$

where $X \in \mathbb{R}^{n \times p}$ is a design matrix containing $p$ features as columns, $y \in \mathbb{R}^n$ is the target vector, and $\lambda > 0$ is a regularization hyperparameter. The Lasso estimator was popularized for variable selection: when $\lambda$ is high enough, many entries in $\theta^*$ are exactly equal to 0. This leads to more interpretable models and reduces overfitting compared to the least-squares estimator.

Solvers for Problem (3) have evolved since its introduction by Tibshirani (1996). After generic quadratic program solvers, new dedicated solvers were proposed based on iterative reweighted least-squares (IRLS) (Grandvalet, 1998), followed by LARS (Efron et al., 2004), a homotopy method computing the full Lasso path[3]. The LARS solver helped popularize the Lasso, yet the algorithm suffers from stability issues and can be very slow for worst case situations (Mairal and Yu, 2012). General purpose solvers became popular for Lasso-type problems with the introduction of the iterative soft thresholding algorithm (ISTA, Daubechies et al. 2004), an instance of forward-backward splitting (Combettes and Wajs, 2005). These algorithms became standard in signal and image processing, especially when accelerated (FISTA, Beck and Teboulle 2009).

In parallel, proximal coordinate descent has proven particularly relevant for the Lasso in statistics. Early theoretical results were proved by Tseng (1993) and Sardy et al. (2000), before it became the standard solver of the widely distributed packages `glmnet` in R and `scikit-learn` in Python. For further improvements, some solvers exploit the sparsity of $\theta^*$, trying to identify its support to reduce the problem size. Best performing variants of this scheme are screening rules (*e.g.,* El Ghaoui et al., 2012; Bonnefoy et al., 2015; Ndiaye et al., 2017) and working/active sets (*e.g.,* Johnson and Guestrin 2015; Massias et al. 2018), including strong rules (Tibshirani et al., 2012).

While reviews of Lasso solvers have already been performed (Bach et al., 2012, Sec. 8.1), they are limited to certain implementation and design choices, but also naturally lack comparisons with more recent solvers and modern hardware, hence drawing biased conclusions.

The code for the benchmark is available at https://github.com/benchopt/benchmark_lasso/. Results obtained on 4 datasets, with 9 standard packages and some custom reimplementations, possibly leveraging GPU hardware, and 17 different solvers written in Python/numba/Cython, R, Julia or C++ (Table E.1) are presented in Figure 4. All solvers use efficient numerical implementations, possibly leveraging calls to `BLAS`, precompiled code in `Cython` or just-in-time compilation with `numba`.

The different parameters influencing the setup are

- the regularization strength $\lambda$, controlling the sparsity of the solution, parameterized as a fraction of $\lambda_{\max} = \left\| X^\top y \right\|_\infty$ (the minimal hyperparameter such that $\theta^* = 0$),

---

[3]The Lasso path is the set of solutions of Problem (3) as $\lambda$ varies in $(0, \infty)$.

- the dataset dimensions: *MEG* has small $n$ and medium $p$; *rcv1.binary* has medium $n$ and $p$; *news20.binary* has medium $n$ and very large $p$ while *MillionSong* has very large $n$ and small $p$ (Table E.2).

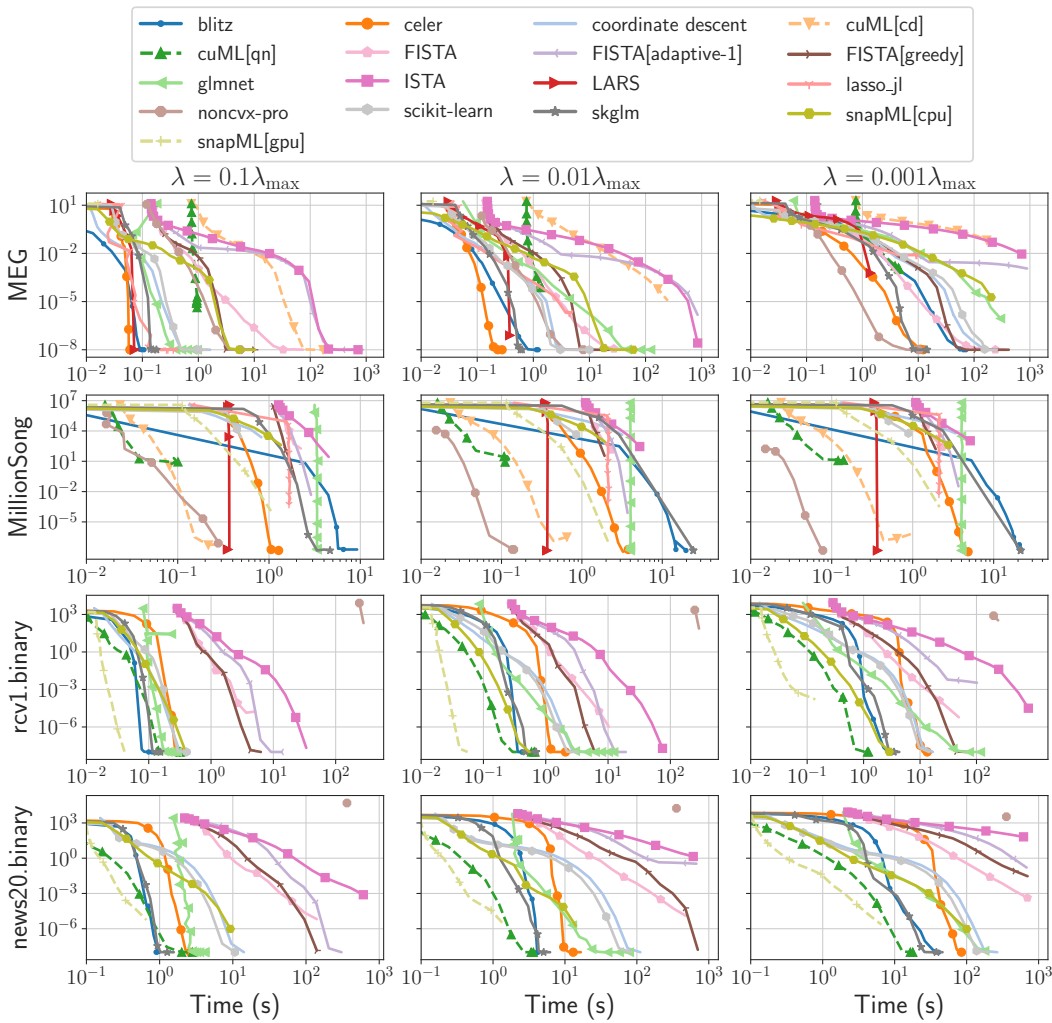

Figure 4: Benchmark for the Lasso, on 17 solvers, 4 datasets (*rows*), and 3 variants of the `Objective` (*columns*) with decreasing regularization $\lambda$. The curves display the suboptimality of the objective function, $f(\theta^t) - f(\theta^*)$, as a function of time.

**Results** Figure 4 presents the result of the benchmark on the Lasso, in terms of objective suboptimality $f(\theta^t) - f(\theta^*)$ as a function of time.

Similarly to Section 3, the GPU solvers obtain good performances in most settings, but their advantage is less clear. A consistent finding across all settings is that coordinate descent-based methods outperform full gradient ones (ISTA and FISTA, even restarted), and are improved by the use of working set strategies (`blitz`, `celer`, `skglm`, `glmnet`). This observation is even more pronounced when the regularization parameter is large, as the solution is sparser.

When observing the influence of the dataset dimensions, we observe 3 regimes. When $n$ is small (*MEG*), the support of the solution is small and coordinate descent, LARS and `noncvx-pro` perform the best. When $n$ is much larger than $p$ (*MillionSong*), `noncvx-pro` clearly outperforms other solvers, and working set methods prove useless. Finally, when $n$ and $p$ are large (*rcv1.binary*, *news20.binary*), CD and working sets vastly outperforms the rest while `noncvx-pro` fails, as it requires solving a linear system of size $\min(n, p)$. We note that this setting was not tested in the original experiment of Poon and Peyré (2021), which highlights the need for extensive, standard experimental setups.

When the support of the solution is small (either small $\lambda$, either small $n$ since the Lasso solution has at most $n$ nonzero coefficients), LARS is a competitive algorithm. We expect this to degrade when $n$ increases, but as the LARS solver in `scikit-learn` does not support sparse design matrices we could not include it for *news20.binary* and *rcv1.binary*.

This benchmark is the first to evaluate solvers across languages, showing the competitive behavior of `lasso.jl` and `glmnet` compared to `Python` solvers. Both solvers have a large initialization time, and then converge very fast. To ensure that the benchmark is fair, even though the `Benchopt` library is implemented in `Python`, we made sure to ignore conversion overhead, as well as just-in-time compilation cost. We also checked the timing's consistency with native calls to the libraries.

Since the Lasso is massively used for it feature selection properties, the speed at which the solvers identify the support of the solution is also an important performance measure. Monitoring this with `Benchopt` is straightforward, and a figure reporting this benchmark is in Appendix E.

## 5   Third example: How standard is a benchmark on ResNet18?

As early successes of deep learning have been focused on computer vision tasks (Krizhevsky et al., 2012), image classification has become a *de facto* standard to validate novel methods in the field. Among the different network architectures, ResNets (He et al., 2016) are extensively used in the community as they provide strong and versatile baselines (Xie et al., 2017; Tan and Le, 2019; Dosovitskiy et al., 2021; Brock et al., 2021; Liu et al., 2022). While many papers present results with such model on classical datasets, with sometimes extensive ablation studies (He et al., 2019; Wightman et al., 2021; Bello et al., 2021; Schmidt et al., 2021), the lack of standardized codebase and missing implementation details makes it hard to replicate their results.

The code for the benchmark is available at https://github.com/benchopt/benchmark_resnet_classif/. We provide a cross-dataset –*SVHN*, Netzer et al. (2011); *MNIST*, LeCun et al. (2010) and *CIFAR-10*, Krizhevsky (2009)– and cross-framework –*TensorFlow/Keras*, Abadi et al. (2015) and Chollet et al. (2015); PyTorch, Paszke et al. (2019)– evaluation of the training strategies for image classification with ResNet18 (see Appendix F for details on architecture and datasets). We train the network by minimizing the cross entropy loss relatively to the weights $\theta$ of the model. Contrary to logistic regression and the Lasso, this problem is non-convex due to the non-linearity of the model $f_\theta$. Another notable difference is that we report the evolution of the test error rather than the training loss.

Because we chose to monitor the test loss, the `Solvers` are defined as the combination of an optimization algorithm, its hyperparameters, the learning rate (LR) and weight decay schedules, and the data augmentation strategy. This is in contrast to a case where we would monitor the train loss, and therefore make the LR and weight decay schedules, as well as the data augmentation policy, part of the objective. We focus on 2 standard methods: stochastic gradient descent (SGD) with momentum and Adam (Kingma and Ba, 2015), as well as a more recently published one: Lookahead (Zhang et al., 2019). The LR schedules are chosen among fixed LR, step LR[4], and cosine annealing (Loshchilov and Hutter, 2017). We also consider decoupled weight decay for Adam (Loshchilov and Hutter, 2019), and coupled weight decay (*i.e.,* $\ell_2$-regularization) for SGD. Regarding data augmentation, we use random cropping for all datasets and add horizontal flipping only for *CIFAR-10*, as the digits datasets do not exhibit a mirror symmetry. We detail the remaining hyperparameters in Table F.2, and discuss their selection as well as their sensitivity in Appendix F.

**Aligning cross-framework implementations**   Due to some design choices, components with the same name in the different frameworks do not have the same behavior. For instance, when it comes to applying weight decay, `PyTorch`'s SGD uses coupled weight decay, while in `TensorFlow/Keras` weight decay always refers to decoupled weight decay. These two methods lead to significantly different performance and it is not straightforward to apply coupled weight decay in a post-hoc manner in `TensorFlow/Keras` (see further details in Section F.3). We conducted an extensive effort to align the networks implementation in different frameworks using unit testing to make the conclusions of our benchmarks independent of the chosen framework. We found additional significant differences (reported in Table F.3) in the initialization, the batch normalization, the convolutional layers and the weight decay scaling.

---

[4]decreasing the learning rate by a factor 10 at mid-training, and again at 3/4 of the training

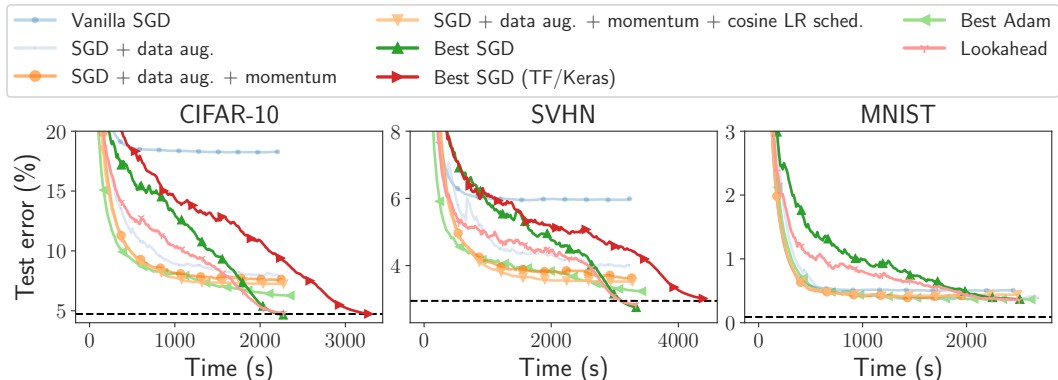

Figure 5: **ResNet18 image classification benchmark with `PyTorch` Solvers.** The best SGD configuration features data augmentation, momentum, cosine learning rate schedule and weight decay. In dashed black is the state of the art for the corresponding datasets with a ResNet18 measured by Zhang et al. (2019) for *CIFAR-10*, by Zheng et al. (2021) for *SVHN* with a PreAct ResNet18, by PapersWithCode for *MNIST* with all networks considered. Off-the-shelf ResNet implementations in `TensorFlow/Keras` do not support images smaller than $32 \times 32$ and is hence not shown for *MNIST*. Curves are exponentially smoothed.

**Results** The results of the benchmark are reported in Figure 5. Each graph reports the test error relative to time, with an ablation study on the solvers' parameters. Note that we only report selected settings for clarity but that we run every possible combination.[5]

Firstly, reaching the state of the art for a vanilla ResNet18 is not straightforward. On the popular website Papers with code it has been so far underestimated. It can achieve 4.45% and 2.65% test error rates on *CIFAR-10* and *SVHN* respectively (compared to 4.73% and 2.95% – for a PreAct ResNet18 – before that). Our ablation study shows that a variety of techniques is required to reach it. The most significant one is an appropriate data augmentation strategy, which lowers the error rate on *CIFAR-10* from about 18% to about 8%. The second most important one is weight decay, but it has to be used in combination with a proper LR schedule, as well as momentum. While these techniques are not novel, they are regularly overlooked in baselines, resulting in underestimation of their performance level.

This reproducible benchmark not only allows a researcher to get a clear understanding of how to achieve the best performances for this model and datasets, but also provides a way to reproduce and extend these performances. In particular, we also include in this benchmark the original implementation of Lookahead (Zhang et al., 2019). We confirm that it slightly accelerates the convergence of the Best SGD, even with a cosine LR schedule – a setting that had not been studied in the original paper.

Our benchmark also evaluates the relative computational performances of the different frameworks. We observe that `PyTorch-Lightning` is significantly slower than the other frameworks we tested, in large part due to their callbacks API. We also notice that our `TensorFlow/Keras` implementation is significantly slower ($\approx 28\%$) than the `PyTorch` ones, despite following the best practices and our profiling efforts. Note that we do not imply that `TensorFlow` is intrinsically slower than `PyTorch`, but a community effort is needed to ensure that the benchmark performances are framework-agnostic.

A recurrent criticism of such benchmarks is that only the best test error is reported. In Figure 6, we measure the effect of using a train-validation-test split, by keeping a fraction of the training set as a validation set. The splits we use are detailed in Table F.1. Our finding is that the results of the ablation study do not change significantly when using such procedure, even though their validity is reinforced by the use of multiple trainings. Yet, a possible limitation of our findings is that some of the hyperparameters we used for our study, coming from the PyTorch-CIFAR GitHub repository, may have been tuned while looking at the test set.

---

[5]The results are available online as a user-friendly interactive HTML file.

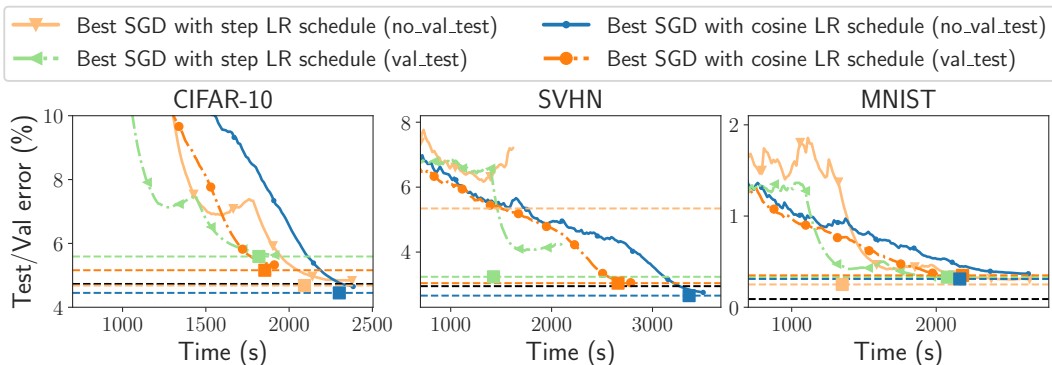

Figure 6: **ResNet18 image classification benchmark with a validation split.** In dashed black is the state of the art (see caption of Figure 5 for more details). In addition, we show in colored horizontal dashed lines, the test results for early stopping on the validation and on the test set for the different solvers, the square mark indicating the moment this stopping would happen. The curves for the train-val splits show the exponentially smoothed median results for five different random seeds.

## 6   Conclusion and future work

We have introduced Benchopt, a library that makes it easy to collaboratively develop fair and extensive benchmarks of optimization algorithms, which can then be seamlessly published, reproduced, and extended. In the future, we plan on supporting the creation of new benchmarks, that could become the standards the community builds on. This work is part of a wider effort to improve reproducibility of machine learning results. It aims to contribute to raising the standard of numerical validation for optimization, which is pervasive in the statistics and ML community as well as for the experimental sciences that rely more and more on these tools for research.

## 7   Acknowledgements

It can not be stressed enough how much the Benchopt library relies on contributions from the community and in particular the Python open source ecosystem. In particular, it could not exist without the libraries mentioned in Appendix A.

This work was granted access to the HPC resources of IDRIS under the allocation 2022-AD011011172R2 and 2022-AD011013570 made by GENCI, which was used to run all the benchmarks. MM also gratefully acknowledges the support of the Centre Blaise Pascal's IT test platform at ENS de Lyon (Lyon, France) for Machine Learning facilities. The platform operates the SIDUS solution (Quemener and Corvellec, 2013).

TL, CFD and JS contributions were supported by the Chaire IA CaMeLOt (ANR-20-CHIA-0001-01). AG, EL and TM contributions were supported by the Chaire IA ANR BrAIN (ANR-20-CHIA-0016). BMa contributions were supported by a grant from Digiteo France. MD contributions were supported by a public grant overseen by the French National Research Agency (ANR) through the program UDOPIA, project funded by the ANR-20-THIA-0013-01 and DATAIA convergence institute (ANR-17-CONV-0003). BN work was supported by the Télécom Paris's Chaire DSAIDIS (Data Science & Artificial Intelligence for Digitalized Industry Services). BMo contributions were supported by a grant from the Labex MILYON.

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
