# OpenReview forum: "Benchopt: Reproducible, efficient and collaborative optimization benchmarks"
_NeurIPS.cc/2022/Conference — NeurIPS 2022 Accept_

### Official Review · Reviewer_Fv41 · 2022-07-08

**Rating:** 4
**Confidence:** 3
**Soundness:** 2 fair
**Presentation:** 3 good
**Contribution:** 2 fair

**Summary:**

This paper introduced Benchopt, a collaborative benchmark suite for optimization algorithms in machine learning. The functionality is showcased on three experimental settings.

**Questions:**

My questions are embedded in the Strengths/Weaknesses Section above.

**Limitations:**

Since no new methodology has been proposed, I don't see a need for extensive discussion of limitations. Some of the weaknesses listed above constitute (in my opinion) limitations of the benchmark itself. I would encourage the authors to comment on them in the rebuttal and possibly add explanations/clarifications to the paper.

**Strengths And Weaknesses:**

### Strengths

1. I commend the effort to improve transparency and reproducibility in OptML research. This is much needed!

2. Designing the Benchmark as a collaborative effort from the start is definitely great. I think the design of the benchmark will make it very easy for researchers to contribute.

### Weaknesses

1. I think it is a bad choice to report comparisons in terms of wall-clock-time. This can be extremely dependent on specific combinations of framework, implementation and hardware. Since the purpose of the BenchOpt is supposedly to compare optimization methods and not specific implementations, I would try to abstract away as much as possible these dependencies. One possibility would be to perform comparisons in terms of number of iterates and report (estimated) time per iteration alongside (or group algorithms roughly according to ther per-iteration complexity).

2. A key quality of optimization methods, at least in deep learning, is ease of hyperparameter tuning. From the main text of the paper, it is not clear what hyperparameter tuning protocol has been employed. Hyperparameter sensitivities should definitely be part of the "output" of a benchmark suite like the one proposed here. Are there any plans for this?

3. On thing that irritates me is that the choice of the weight decay parameter as well as data augmentation techniques are considered as part of the optimization method. In principle, these are part of the optimization objective. What led the authors to consider this part of the optimizer instead?

---

> ### Author Response · Authors · 2022-08-01
> **Answer to reviewer Fv41**
>
> We thank the reviewer for the comments and we address specific points below:
>
> 1. **Wall-clock-time vs iteration:** Measuring time or iteration are two alternatives that make sense in their respective contexts. Practitioners mostly care about the time it takes to solve their problem, while researchers in mathematical optimization may want to abstract away the implementation and hardware details and only consider iteration. The benchmarks we have presented showcase efficient implementations and are also interested in hardware  and implementation differences (e.g. CPU vs GPU solvers for the Lasso, torch vs TF for the resnet), hence our focus on time.
> However, Benchopt does not impose a choice between the two measures: it is perfectly possible to create plots as a function of the number of iterations as now evidenced in Appendix D.3. Once again, Benchopt aims at providing a versatile tool for its users, not to impose strong design choices.
>
> 2. **Hyper-parameter tuning** In the presented benchmarks, most hyper-parameters have been set to default values or the solver has been run for multiple values on a grid. Specifically for the ResNet benchmark, we describe in section E.1 how we chose the hyperparameters of the network.
> An important part of the design of Benchopt is that a given solver can easily be run with multiple values of its hyper-parameters, as shown in Appendix B.3. It is thus straightforward to run a solver with different sets of hyper-parameters to assess its sensitivity. We also would like to add tools to ease such sensitivity analysis in the future.
> Finally, adding benchmarks on hyperparameter tuning is naturally within the scope of future works for Benchopt.
>
> 3. **Optimizer definition including DA and WD:** In a benchmark concerned with minimizing the train loss, indeed weight decay (WD) and data augmentation (DA) would be part of the Objective.
> In the considered benchmark, *we choose to monitor the test loss*. In this setting, DA and WD are key parts of the optimization algorithm as they are meant to decrease the test loss. One can therefore think of the combination of Optimizer + DA + WD as the algorithm to minimize the test loss. Finally, note that in Pytorch, WD is explicitly made part of the optimizer and not of the loss function, so the fact that WD should be part of the cost function is not unanimous.

---

> > ### Comment · Reviewer_Fv41 · 2022-08-09
> > **Thanks for the reply**
> >
> > Thanks for your response.
> >
> > **Hyperparameter tuning:** It is great that the library can support evaluation of hyperparameter sensitivity. But understand that I am not reviewing a library but the present *paper*, in which you ignore the topic of hyperparameter sensitivity and are opaque about how hyperparameters were chosen in the presented experiments. Since the ease of hyperparameter tuning is a key quality of optimization algorithms, this constitutes a major shortcoming in my opinion.

---

> > > ### Author Response · Authors · 2022-08-09
> > > **Re: thanks for the reply**
> > >
> > > We fully agree that if finding good hyperparameters turns out to be difficult in practice, it seriously limits the potential usefulness of a method.
> > >
> > > For the non-deep learning benchmarks we typically use the textbook parameters for step sizes (e.g. 1/L stepsize for the gradient-based methods in Lasso and Logistic regression) and their estimation is taken into account in the benchmark results as it is part of the run time.
> > >
> > > For the deep learning benchmark we aimed at replicating the documented and published state of the art, taking parameters from reference open source implementations, and we will make this clearer in the final version of the paper. We also performed an ablation study with cross validation (Figure 6) and have shown that the selection of the best optimization strategy was not sensitive to the usual practice of not using a validation set. Moreover, as we agree that the impact of hyper-parameters is of interest, **we will add in the final camera ready version of this paper a grid search to evaluate the sensitivity of the algorithms to parameter selection and we will display the performance using parallel coordinates plot**. In this extra experiment, we will evaluate the impact of selecting the learning rate, momentum and weight decay. We will also stress the importance of evaluating the hyperparameter sensitivity for optimization benchmarks in general.
> > >
> > > Now regarding the scientific positioning of the present paper, we would like to point out that it overlaps with two main topics in the NeurIPS call for papers (https://nips.cc/Conferences/2022/CallForPapers): *Optimization (e.g., convex and non-convex optimization)* and *Infrastructure (e.g., datasets, competitions, implementations, libraries)*. With this in mind, we would like to argue that what we contribute here is also a library, as that it fits the scope of NeurIPS. (See also past very successful examples such as PyTorch NeurIPS 2019,  PyGlove NeurIPS 2020 or POT JMLR 2021 ) With the presented benchmarks we aimed to demonstrate that the benchopt software offers the necessary machinery to **impact the science that is produced in the ML optimization community** and, as explained in the paper, the benchmark code is meant to be maintained and extended by a community effort in the coming years.

---

### Official Review · Reviewer_4Y29 · 2022-07-10

**Rating:** 7
**Confidence:** 4
**Soundness:** 4 excellent
**Presentation:** 3 good
**Contribution:** 3 good

**Summary:**

A python-based toolbox for benchmarking optimization methods, with special focus on machine learning applications, is presented in this paper. The modular design of the toolbox allows a flexible assessment, to account for the impact that rapid development of software and hardware has in the field. Moreover, although the benchmarking-suite is written in python, it allows the comparison of methods written in other programming languages such as julia. To exemplify the usage of this toolbox, the paper compares several methods for three widely-known problems in the machine learning and deep learning community (logistic regression with l2 regularization, Lasso and the paradigmatic case of ResNet). This case study illustrates the benefits of the proposed approach and manage to make some novel discoveries.


**Questions:**

- How does the toolbox determine that an algorithm has converged? Does this criteria vary with the type of problem (L2LogReg, lasso, ResNet, etc.)?

- Regarding the alignment of TensorFlow and PyTorch for ResNet18 training, does the toolbox offer some built-in way of simplifying this process or just a series of recommendations on how to do it? Is it possible that two different researchers deal with the alignment in two different ways when comparing TensorFlow and PyTorch implementations, rendering the comparisons invalid?
I believe that prospective users could benefit from a way of simplifying the process of “alignment” between those two frameworks that is as transparent as possible to the user, and not just guidelines.  For example, is it possible to implement “adaptive layers” to be added to the network architecture with the purpose of dealing with some of the aspects described by authors in section E.2? This would leave less important details of the benchmarking process free to the user.


**Limitations:**

I believe authors should mention other toolboxes with similar purpose and explicitly mention the differences between their approach and the currently available tools.

To my mind, authors could also discuss more about the limitations of the alignment process between frameworks.

**Strengths And Weaknesses:**

Positive remarks:
- The modular design of the benchmark guarantees the versatility that such a tool requires to meet the community needs.
- Interesting findings for future research (that is what benchmarks are supposed to be used for!):
	- For the l2-regularized logistic regression, recommendations on the use of larger datasets to explore the stochastic solvers.
	- For Lasso, assessments include julia-based methods for the first time in a systematic comparison. Additionally, other objectives, like the sparsity of the solution, can be easily analyzed as well.
	- For ResNet 18 training, it is shown that reaching the state-of-the-art for this architecture is more dependent on parameters such as the data augmentation strategy and weight decay/learning rate adjustments.

The originality of the article is limited by the fact that there are similar approaches within this area of research. Nevertheless authors show an interesting proposal of a toolbox mainly oriented to the ML community. The presentation of the proposed toolbox is acceptable, but there are some points that, to my view, require clarification.  I believe that the analysis of the state-of-the-art of optimization benchmarks should include similar toolboxes like MLPerf, COCO [1], Nevergrad [2] or FitBenchmarking [3], and explicitly describe the differences between this toolbox and those approaches. Otherwise, authors risk not highlighting the significance of their work.

[1]Hansen, Nikolaus, et al. "COCO: A platform for comparing continuous optimizers in a black-box setting." Optimization Methods and Software 36.1 (2021): 114-144.

[2] https://github.com/facebookresearch/nevergrad

[3] Markvardsen et al., (2021). FitBenchmarking: an open source Python package comparing data fitting software. Journal of Open Source Software, 6(62), 3127,

---

> ### Author Response · Authors · 2022-08-01
> **Answer to reviewer 4Y29**
>
> We thank the reviewer for the encouraging evaluation and constructive comments. We address the points that were raised in the review below.
>
> **Originality of the framework** We have clarified in our general comment the originality of Benchopt compared to other similar initiatives. We will add references to them in the intro in the final version of the manuscript which will include an extra page. We have now added references to COCO and nevergrad in a new Appendix I that presents a new benchmark of zero-order optimizers applied to classifical non-convex functions. Doing so, we demonstrate that Benchopt can be easily used to evaluate black-box zero-order optimizers across packages. Indeed this novel benchmark compares 5 optimizers from the nevergrad package, but also 3 optimizers from optuna package as well as some solvers in scipy.optimize. While Benchopt is not as extensive for zero-order evaluation as COCO, this showcases that both frameworks pursue similar goals, yet Benchopt aims to be particularly relevant for machine learning use cases.
>
> **Measure of algorithm convergence**: Benchopt offers several ways to measure this. The solver can be stopped when the objective value does not decrease significantly between iterations. For convex problems, we also propose to track the duality gap (which upper bounds the suboptimality $f(\theta^t) - f(\theta^*)$), as is done for the Lasso.
> For non convex problems, criteria such as gradient norm or violation of first order conditions can be used, as users do in practice. These criteria can easily be customized.
>
> **Model alignment between PyTorch and TF:** To answer briefly: there is no way/tool to automate the alignment of models implemented in the 2 different frameworks at this stage. It is also not the aim of Benchopt to provide such a tool. A potential candidate library for this would be the ONNX library but for now, it only ensures that a model gives the same predictions at evaluation time in two frameworks, not during the training.
> For each new benchmark/framework, the alignment needs to be done for the considered model but it only needs to be done once for the whole benchmark.
> Our intent with section E.2. was to show that such alignment work would benefit greatly from being mutualized in the community. There is not an infinite number of basic test cases for deep learning optimizers, and having them aligned once and for all in Benchopt benchmark instances would greatly simplify DL optimization research across frameworks.

---

### Official Review · Reviewer_nYXq · 2022-07-17

**Rating:** 7
**Confidence:** 4
**Soundness:** 3 good
**Presentation:** 3 good
**Contribution:** 3 good

**Summary:**

The authors provide and describe a library, Benchopt, which can be used to benchmark optimizers.  The authors demonstrate and discuss the results of three benchmark models on a number of tasks: l_2 regularized logistic regression, Lasso, and ResNet-18.

**Questions:**

- Could you explain in more detail the design decisions underlying the library?  How should it be used?  Does the library make strong design choices as to how a set of optimizers should be evaluated?

- Can you discuss further in detail how a researcher using Benchopt should deal with the possibility that one could draw different conclusions when comparing methods based on how the benchmarking experiments are set up?  Similarly, how researchers can evaluate optimizers for multiple, possibly opposing, criteria such as accuracy and efficiency/computational cost?

- You note that implementations across libraries may have notable differences in how the optimizer is designed (e.g. Pytorch vs Tensorflow).  Who's responsibility in the Benchopt ecosystem is it to deal with these issues? The Benchopt maintainers? Contributors? Researcher users?

**Limitations:**

See the discussion of the weaknesses in the limitations.  In particular, the technical and methodological difficulties regarding how one draws conclusions when comparing optimizers could be further emphasized, as conclusions are not always clear and can be contradictory.  The citations provided in the weaknesses section are some of the relevant prior work in this area.

**Strengths And Weaknesses:**

Strengths:

- The results and methods are clearly presented.  The paper overall is easy to read and understand.

- The authors present a broad set of experiments across a variety of tasks, demonstrating the applicability of the library.

- The authors identify an important problem in machine learning, the benchmarking and comparisons of optimizers.

- The authors provide a good discussion of related work.

Weaknesses:

- The organization of the paper could be improved.  The discussion of the results and the specific experiments, to me, are not dramatically different compared to similar optimizer benchmarking papers e.g.:

https://arxiv.org/abs/2206.05985

https://proceedings.neurips.cc/paper/2021/hash/17fafe5f6ce2f1904eb09d2e80a4cbf6-Abstract.html

https://arxiv.org/abs/1910.05446

https://ieeexplore.ieee.org/document/9138989

http://proceedings.mlr.press/v139/schmidt21a.html

http://proceedings.mlr.press/v119/sivaprasad20a.html

As a result, the presentation and discussion of the results alone are not sufficient to demonstrate the utility of the library.  In my opinion, section B in the appendix provides further technical details about the library that suggest its broad applicability and ease of use.  Further emphasis on the specific technical aspects of the library would further strengthen the argument for the uniqueness of Benchopt.

- The experiments specifically used do not address the concerns regarding optimizer benchmarking identified by the authors in the introduction:

1. " there are no clear community guidelines on how to perform, share, and publish benchmarks" - these aspects of benchopt's design are discussed in passing in section 2, but the specific system details and the motivation for these choices is not discussed in detail.  Again, Appendix B provides some insight into these design choices.

2. "many others fall short in this regard due to lack of time and resources, and in turn feature results that are hard to reproduce by other researchers" - The experiments presented, while broad, are relatively computationally cheap.  Moreover, the authors do not make clear why these results are specifically more reproducible than other optimizer benchmarks, as many of these other papers have provided code.  In fact, the limited discussion regarding the inability to reproduce the benchmarks provided by Papers with Code, leaves the reader wondering why that benchmark is not reproducible.

- The authors note that "for practical evaluation, the devil is in the details."  Prior work (including the citations mentioned above) has also emphasized this observation.  However, the authors presentation of comparisons across methods is relatively straightforward, in that the authors often provide one specific way to evaluate a task as opposed to multiple, possibly differing ways of performing evaluation, such as with broader ablations or hyperparameter searches.  Works such as Choi et al, Cooper et al, Daulton et al, Sivaprasad et al, and Schmidt et al have noted that *how* comparisons are made leads to different conclusions.  While the means by which these comparisons are made in Benchopt may be left to the user, this detail could be further emphasized in the methods and discussion.

----
Overall impression:

I think this work is relevant to the NeurIPS community and would be of use.  The weaknesses in the paper are largely aesthetic, as they are not with respect to the library in itself.  I believe these limitations are surmountable, and that the paper would benefit by being addressed.

----

Update post discussion period:

I appreciate the discussion points of the authors and again believe that the adoption of tools such as BenchOpt would improve the level of rigor within the NeurIPS research community.  I am raising my score to an accept to reflect these events.

---

> ### Author Response · Authors · 2022-08-01
> **Answer to reviewer nYXq**
>
> **Specificity of Benchopt with respect to other benchmark papers:** As mentioned in the general answer, the distinguishing feature of benchopt relative to other benchmark frameworks is that it is a meta-benchmark tool. Thus, the core difference with the provided references is that it does not consider a fixed set of problems/datasets, and was designed from the beginning to be evolutive and collaborative. Moreover, most of these papers do not provide code to reproduce, update or extend their results. We have further emphasized this in the core of the manuscript and added more links to appendix B.
>
> **Experiments does not address raised concerns:** We have now better highlighted the design description from Appendix B into the main part of the paper, to clarify our guideline for performing a benchmark. We enforce reproducibility by sharing implementations in centralized repositories per benchmark (as opposed to paperwithcode, where each method is implemented in its own benchmark). The benchmark results can thus be reproduced at once by cloning the benchmark and running it with our CLI. Moreover, Benchopt is not a fixed, limited set of benchmarks, but a framework to create open and evolving benchmarks. It is then straightforward to add another optimizer, dataset or hyperparameter to compare in a few lines of code, and then publish the resulting benchmark online for others to build upon.
> For time and resources, we also referred to the burden of reimplementing baselines and competitors, whereas with Benchopt’s design all competitors are readily available at the same location in an open source way.
>
> **Design decisions:** Benchopt emphasizes modularity of the benchmarks and its main feature is the decomposition into Objective, Solver and Dataset classes. The success metric can differ greatly across benchmarks, hence Benchopt leaves the user free to use standard metrics or to customize the evaluation, for example by designing a new one (support recovery for the Lasso benchmark, generalization loss for resnets, etc.) and allows to compute/display together several metrics that can be complementary. We aim at providing researchers with a flexible tool, hence have tried to make as few strong choices as possible, leaving these decisions to each benchmark’s community.
>
> **Interpretations & tradeoff** The goal of Benchopt is to provide a clear and complete picture of the behaviors of algorithms in a wide variety of settings. The interpretation of this picture, on the other hand, remains left to the researcher, as there is no single best rule depending on the problem or application. Once again, these decisions are to be done independently for each benchmark, or even based on the considered task (Resnet uses test error, Lasso uses support recovery and loss, etc.).
>
> **Implementation discrepancy:** In many benchmarks, there are several implementations of the same algorithm that can differ -- *e.g.* CD for the Lasso benchmark. Benchopt allows evaluating how these discrepancies affect the performances, which is an important goal for a benchmark. The specific implementation that is used for each solver should be clear in the benchmark and we think this is the responsibility of the researcher that publishes it, for instance using tables as the ones in our appendix. As the solvers are open source it is also easy to look at the code to see what language or framework is used for each implementation (Numba, Cython, C++, Julia etc.). This is also a key point in our TF/torch alignment, showing implementation details which make the frameworks different can matter, *e.g.* coupled/decoupled weight decay. Also note that a large part of the alignment was to ensure that the different model implementations were the same. This is a necessary part of the benchmark (making sure things are comparable) and making it centralized at the benchmark level will make it less error prone.

---

### Author Response · Authors · 2022-08-01
**General comment to all reviewers and area chairs**

We thank the reviewers for their nice comments about the library and for considering that this paper “*identify an important problem in machine learning*” and that such tool “*is much needed*”. From our understanding, the key points raised by the reviewers are not about the library itself but more about the positioning of the paper and the presented benchmarks.

**Position of benchopt:** We thank the reviewers for providing useful references and we will include them in the final version that includes an extra page. This will allow us to better position our work relative to these other tools. A distinctive feature of benchopt is that it is **a meta benchmarking toolbox**. We do not provide a fixed set of cost functions but provide a recipe to create, maintain and run benchmarks in an extendable and collaborative way. On the contrary, the provided references are each dedicated to a limited and fixed set of problems. We highlight the key differences with related benchmark tools:
In paperswithcode, each repository corresponds to a single method, while in Benchopt all solvers for the same problems are harmonized and shared in the same repository and can be discussed and compared in a single location. Each Benchopt repository comes with continuous integration running everyday, ensuring that code remains runnable.
ML perf is targeted for hardware providers to showcase chip performances on ML tasks and not algorithms. It is designed as competitions, where a fixed set of problems are to be solved and the evaluation is in terms of FLOPS.
Nevergrad is a library dedicated to derivative-free optimization. It offers a number of optimizers that Benchopt can compare against other zero-order optimizers (e.g. optuna) as now showcased in a new benchmark in Appendix I.
Fitbenchmarking is dedicated to a single problem, non-linear least squares. In their formulation, a benchmark corresponds to a dataset. In comparison, Benchopt aims to *provide researchers with tools* so that they can create their own benchmark easily or contribute to the ones from the community.
Other references provided were mostly benchmark papers who proposed a fixed set of problems and methods to evaluate one given tasks.

Ideally, our tool could be used to assess the performance of a new optimization algorithm in a reviewing context: the state-of-the-art solver being available and open source in the library, adding a single new competitor has a low barrier for the authors (who would avoid reimplementing all baselines from scratch).


**Presented benchmarks:** The list of benchmarks we proposed in the paper is not meant to be definitive, and we hope the list will grow thanks to the community (we propose a template for new users to easily add relevant optimization problems).
Each showcases the effectiveness of Benchopt to create benchmarks but also provides interesting finding on the considered problems, as mentioned by reviewer **4Y29**. Moreover, each proposed benchmark is also meant to evolve, as our goal is for the community to continuously extend and improve them in a collaborative effort.
To address some of the concerns raised by the reviewers we have added two additional benchmarks.
The first one (cf. appendix Figure D.2 in the revised document) is aiming to address one concern from reviewer **Fv41**: it proposes a comparison on the Lasso for FISTA, CD and ISTA in terms of iterations instead of wall-clock timing (1 iteration = 1 pass over the full dataset here). The second one is on *zero-order optimization*. It shows that one use case of Benchopt can be to perform similar comparisons as the one in COCO.

---

### Author Response · Authors · 2022-08-08
**DIscussion before end of reviewers-authors exchange period**

Dear Reviewers,

We thank you again for your feedback on our work.
We hope that our revised manuscript, our extra experiments and our clarification have answered your concerns.
Please let us know of any further questions you have before the end of the discussion in one day.

The authors

---

### Meta-Review · Area_Chair_ZcGu · 2022-08-26

**Recommendation:** Accept
**Confidence:** Less certain

**Metareview:**

I thank the reviewers and authors for their work throughout the reviewing process, including the detailed discussion. The paper was borderline after the first round of reviews, but the discussions and my own look at the paper makes the balance tip over to acceptance.

I encourage the authors to use all information in the discussion threads to improve their manuscript. In particular, I agree with Reviewer Fv41 that the main weakness of the paper is the treatment of hyperparameters. That being said, I understand that comparing optimization algorithms for fixed (or a small set of) hyperparameters is already a useful task for the community, and already a difficult one to automatize. I add a few comments below to the reviews already provided, hoping to see them addressed in the camera-ready version.

### Major comments
* how do you report traces for randomized algorithms, such as SGD? I don't see error bars on the traceplots, are there no repetitions? Did you not observe variance in the traces for different seeds? Even for a deterministic algorithm, one typically randomizes, say, the starting point, especially on nonconvex problems.
* In the same vein, if a solver means an algorithm and its hyperparameters, there might be a large number of solvers if many people contribute to a benchmark. If the output of each solver is random, ranking the means of all outputs is a parallel statistical testing problem. How do you propose to treat it? I agree that this is a problem for most benchmarks of randomized algorithms, but one that I rarely saw addressed in a systematic manner.

### Minor comments
L53: automatic
L273: solver's?
L274: all possible combinations
L275: material
L1052: what is "on standard"?

**Award:**

No

---

### Decision · Program_Chairs · 2022-09-14

Accept